# Development of Simple Method for Flood Control Capacity Estimation of Dam in South Korea



Heechan Han [1] , Jaewon Kwak [2], Deokhwan Kim [3,*] , Jaewon Jung [3] , Hongjun Joo [4] and Hung Soo Kim [5]

1   Blackland Research and Extension Center, Texas A&M AgriLife, Temple, TX 76502, USA; heechan.han@ag.tamu.edu
2   Han River Flood Control Office, Ministry of Environment, Seoul 06501, Korea; firstsword@korea.kr
3   Department of Hydro Science and Engineering Research, Korea Institute of Civil Engineering and Building Technology, Goyang 10223, Korea; jaewonjung@kict.re.kr
4   Department of Construction Certification Center, Korea Institute of Civil Engineering and Building Technology, Goyang 10223, Korea; engineer1026@kict.re.kr
5   Department of Civil Engineering, Inha University, Incheon 22212, Korea; sookim@inha.ac.kr
*   Correspondence: kimdeokhwan@kict.re.kr

**Abstract:** As flood damage is becoming more frequent and severe worldwide, efficient flood control of dams has been highlighted. The purpose of the study is to establish a simple method for dam operators to estimate the flood control capacity and predischarge level required for flood response. The cumulative probability distribution (CDF) pair with the same return period for 12 major dams located in South Korea were estimated using the frequency matching method. A Ratio of Storage volume to Flood inflow (RSF) concept was suggested and applied for each dam, and they were classified into three types: Linear, Estranged, and Vague according to the water storage characteristics. Using the method presented in this study, we suggested the required amount of flood control capacity and target water level for each dam. The results demonstrated that there is no linear relationship between flood and storage of dam when the ratio of watershed area to a storage capacity of the dam is 2.0 or more, or the ratio of watershed area to flood control capacity is 20.0 or more. Further, it was found that the RSF value is affected by the initial water level of the dam when a high flood inflow was observed for Estranged and Vague types. It is expected that the method presented in this study can be basic information for performing predischarge for flood control in numerous dams.

**Keywords:** frequency matching method; flood control capacity; predischarge of dam

## 1. Introduction

Natural disasters have become more severe and have doubled during the past 20 years [1]. For example, in August 2020, over 1000 mm of precipitation that caused severe flood damage was observed over South Korea [2]. In Japan, a massive flood occurred with over 600 mm of precipitation for two days and caused 212 deaths [3]. Thus, the role of dams for efficient flood control is becoming more important, and various methods for practical dam operation and decision-making are being proposed.

Since floods have coexisted with civilization in human history, many studies have been conducted on developing methods for effective flood control [4]. Generally, flood response in watersheds can be through dam operation and river management. The flood control of dams is carried out in a way that reduces the amount of flooding by storing the inflowing flood and then releasing it in consideration of the condition of the downstream river. In the case of flood management in river, it is based on forecasting and warning the risk of flooding in consideration of the river level [5]. However, since the flood control capability of dam is affected by several conditions such as water level, storage, inflow, and discharges, the effectiveness of flood response is highly dependent on the skills of experienced experts [6].

The dam operation method for flood control has been generally referred to as the Reservoir Operation Method (hereinafter as ROM). Representative examples of ROM contain Technical ROM, Rigid ROM, Auto ROM, and Spillway rule curve ROM. The ROM aims to reduce the peak flood water level of downstream rivers by storing a part of the flood runoff in the dam upstream [7,8]. Therefore, for effective flood control, estimating an appropriate level of storage for the flood inflow is an important key. General dams have Flood Water Level (hereinafter referred to as 'FWL'), a Normal High-Water Level (hereinafter referred to as 'NHWL'), and a Restricted Water Level (hereinafter referred to as 'RWL') to secure the flood control capacity for reliable flood control [9,10].

However, since flood events with the frequency of 200 to 500 years may occur due to the impact of climate change [11], the flood control capacity of dams, which was designed based on the 200 (or under) year frequency, may be hard to respond properly [10]. To make the flood control robust, it is necessary to increase the flood control capacity of the dam [12]. However, since South Korea relies on dam water supply for 48% of its annual water resource usage, securing an excessive flood control capacity may adversely affect the water supply ability of the dam [13]. Thus, in addition to the flood control capacity secured through NHWL and RWL, additional flood control capacity has been secured and responds through predischarge when heavy rainfall is expected on the watershed [14]. For accurate predischarge, various environmental conditions such as hydrological cycle, dam safety, and flood characteristics in upstream and downstream areas must be comprehensively considered. Further, there is a limitation of dam operation in that it highly depends on the skills of experienced experts and advanced hydrological analysis. This means that an effective and simple method is required for dam operators to easily determine the amount of predischarge in advance.

Therefore, the main objective of this study is to develop a simple method for dam operators to easily estimate the required flood control capacity based on weather forecasts and to implement predischarge when heavy rainfall is expected. For this, the frequency matching method was applied to the daily dam inflow and discharge for 12 major dams located in South Korea, and the cumulative probability distribution (CDF) pair with the same return period was estimated. A Ratio of Storage volume to Flood inflow (RSF) concept was suggested and analyzed for each dam and it is classified into three types: Linear, Estranged, and Vague according to the flood storage characteristics for the dams. Finally, based on the analysis results, the method to estimate the required amount of flood control capacity and target water level for each dam were suggested.

## 2. Materials and Methods

### 2.1. Frequency Matching Method

Frequency matching is a method to match the frequencies of two datasets [15]. It is performed by arranging data independently for two sets of data (i.e., inflow and storage of dam) in ascending (descending) order and tying data with the same frequency into a pair. Sorted pairs of data are less likely to appear in natural conditions, but each datum has the same frequency of return period [16]. This method has been generally applied for systematic error correction or sampling and has been used in various fields due to its simple applicability. In the field of hydrology, it has been used for bias correction of predicted rainfall [17–19] and radar rainfall estimation [20]; it has also been used to obtain Curve Number (CN) in the Natural Resources Conservation Service (NRCS) CN method [21].

The frequency matching method assumes that the output and input events have the same frequency. Similarly, in the rainfall–runoff relationship, it can be assumed that "if the 100-year frequency of rainfall occurred, the resulting flow rate also has the same 100-year frequency" (see Figure 1). The flood control capabilities of dams can be simulated as a series of processes in which the inflow is stored and discharged according to the artificial operation of the dam. This is more direct than the rainfall–runoff relationship used to derive the runoff curve index. Therefore, in this study, the frequency matching method, which considers a concept where "If the maximum inflow to the dam occurs in a specific year, the

resulting dam discharge also corresponds to the maximum discharge", was applied to the series of inflow and outflow of the dam.

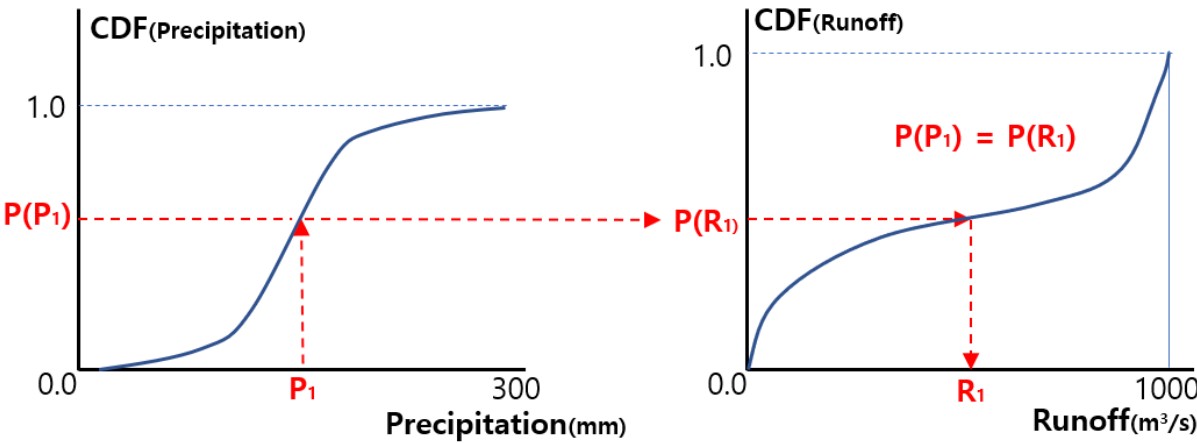

**Figure 1.** Concept of frequency matching in NRCS method.

### 2.2. The Ratio of Storage Volume to Flood Inflow (RSF) for Flood Control of the Dam

In general flood control of dam, the inflow (I) is stored up through an appropriate level, and a stored water is discharged (O) in consideration of the downstream river conditions. Since the dam storage and discharge do not occur at the same time and there is a lag time between both events, the flood control capability of the dam has been analyzed based on how much the peak flood water level can be reduced downstream via the dam operation [22–28].

To overcome these limitations, the frequency matching method is applied to the inflow and outflow series of the dam. Figure 2a illustrates a fundamental concept of flood control of dam. When the frequency matching method is applied to the dam inflow and outflow, they can be expressed as a pair of the same cumulative distribution function (CDF), as shown in Figure 2b. If the dam inflow (I) and discharge (O) volumes correspond to the CDF of the same frequency, it can be free from the problem of time lag between two events, and the amount of dam storage (S = I − O) can be defined as the difference in inflow and outflow (area) of tail of the CDF, as expressed in Equation (1).

$$S = I - O = \int_{R_{I_1}}^{R_{I_2}} RdR - \int_{R_{O_1}}^{R_{O_2}} RdR \tag{1}$$

where, $R_{Ii=1,2}$ are the threshold levels (or criteria) of inflow rate that are recognized as an annual flood event on CDF and $R_{oi=1,2}$ are also thresholds level for outflow CDF. Considering that the discharge volume is generally adjusted according to the inflow from upstream area [10], stored volume of flood (S) can be expressed as the portion ($\alpha$) of flood inflow volume (I) and the RSF ($\alpha$) values can be indicated as Equations (2) and (3). The RSF value can be calculated using the database observed at each dam, and it indicates how much the dam temporarily stores the amount of water flowing into the dam during a flood event.

$$S = \alpha I = I - O \tag{2}$$

$$\alpha = \left(1 - \frac{O}{I}\right) = \left(1 - \frac{\int_{R_{I_1}}^{R_{I_2}} RdR}{\int_{R_{O_1}}^{R_{O_2}} RdR}\right) \tag{3}$$

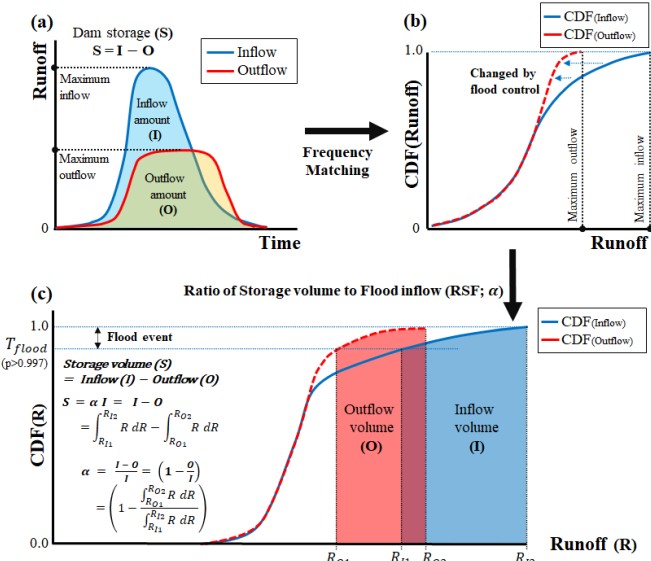

**Figure 2.** The concept of ratio of storage volume to flood inflow (RSF): (**a**) dam operation; (**b**) CDF plot of dam operation with frequency matching; (**c**) RSF formulation.

## 3. Results

### 3.1. Study Area

In this study, twelve major dams located in the major rivers of South Korea were selected to analyze flood control capability using RSF concept. For general analysis of dams, it is desirable to target as many dams as possible; however, considering the availability of datasets, dams with reliable datasets with 20 years or more were selected as target dams in this study. The target twelve dams are as follows: Hwacheon, Soyanggang, Chungju, Hoengseong, Andong, Imha, Hapcheon, Namgang, Milyang, Yongdam, Daecheong, and Sumjingang (Figure 3). The total area of the upstream basin of these dams is about 25,000 km$^2$, which is approximately 25% of the total area of South Korea, and the annual water supply is 10,702.6 million m$^3$ [29]. Historical records of daily dam inflow, storage, discharge, and other features were obtained from the Water Resources Information System [30], the K-Water [31], and the Korea Hydro & Nuclear Power Company [32].

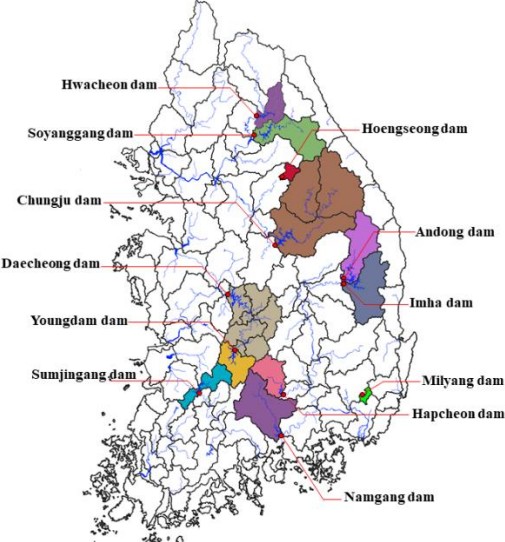

**Figure 3.** Description of study area; red circles indicate the location of each dam and colored areas indicate the upstream area of each dam.

### 3.2. RSF Estimation of Dam

In order to estimate the RSF ($\alpha$) proposed in the Section 2.2, the frequency matching method was applied to the daily inflow and discharge of 12 dams, and CDF pairs of the same frequency with the same return period were derived. To derive RSF values from CDF pairs, the threshold level (or criteria) for separating the flood event is required. In general, multipurpose dams are designed for flood response with a frequency of 200 years [10]. This is the nonexcess probability, which is $p = 0.995$, and the median duration of inflow exceeding the nonexcess probability ($p = 0.995$) in the dam was also estimated as 2 days. Thus, the duration of flood inflow events exceeding the nonexcess probability ($p = 0.995$) are generally considered for 2 days. In addition, according to Lee et al. (2011) [33], the average flood duration is about 2 days (48 h) when designing the dam for reliable flood response. Therefore, if the maximum flood occurs in the dam every year, it can be assumed to have an average duration of 2 days, which corresponds to the nonexcess probability ($p > 0.995$). Based on this concept, this study calculates the storage ratio (RSF) of dam at the time of the annual maximum flood occurrence by using the difference between the inflow and discharge volume corresponding to the nonexcess probability ($p > 0.995$; top 2 days) in the CDF pairs from the 12 target dams. Figure 4 shows the scatter plot of RSF values for each dam.

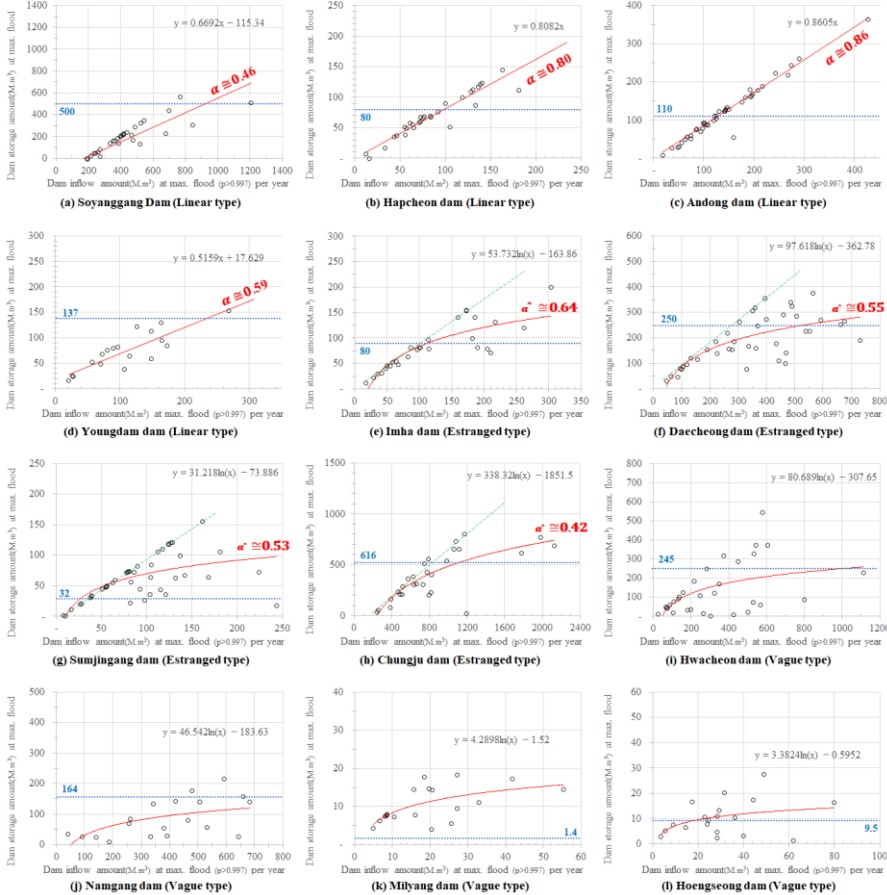

**Figure 4.** Scatter plot of RSF values of each dam for (**a**) Soyanggang, (**b**) Hapcheon, (**c**) Andong, (**d**) Youngdam, (**e**) Imha, (**f**) Daecheong, (**g**) Sumjingang, (**h**) Chungju, (**i**) Hwacheon, (**j**) Namgang, (**k**) Milyang, (**l**) Heongseong; each point indicates RSF value for every year; red line indicates regression line of RSF points; $\alpha$ indicate the RSF value with linear regression line and $\hat{a}$ indicate mean RSF value for logarithmic type; blue dot–lines indicate flood control capacity; green dot–lines denote the high threshold of RSF for (**e**) to (**h**).

*3.3. Method for Estimation of Predischarge of Dam*

In addition to the flood control capability based on NHWL and RWL, it has been secured and responded through predischarge when heavy rainfall is expected on the watershed [14]. However, for reliable estimation of predischarge, various conditions such as hydrological analysis, dam safety, and up- and down-stream conditions should be considered in advance. Moreover, it has the limitations of highly relying on the experience of professional experts and an advanced hydrological analysis system. Thus, RSF ($\alpha$) values presented in this study that represent the ratio of dam storage to the flood inflow volume can be used to simply calculate the amount of inflow that should be stored and discharged as predischarge from the dam.

Figure 5 shows the scatter plots of RSF values for Andong dam. The results show that the RSF value is 0.86 based on the flood inflow and storage data, which demonstrates that the Andong dam controls the flood in the basin by storing about 86% of the flood inflow and discharging only 14% downstream. By applying RSF method, it can be utilized as a quantitative flood response standard method rather than depending on the skills of experts for determining predischarge of dams. For example, when rainfall of 200 mm occurs for two days in the entire basin, the flood volume flowing into Andong dam is about 221.8 million m$^3$ with the runoff rate of 70%. In this case, since the amount of storage that must be stored in the dam to satisfy the RSF value is about 191 million m$^3$, the efficient flood control is possible if predischarge is implemented to store about 191 million m$^3$ based on the RSF value (0.86) of Andong dam. Although the amount of predischarge of each dam can be estimated through advanced hydrological analysis method, considering the uncertainty of rainfall forecasting and the need for a method to determine predischarge easily [34], the RSF method is expected to be applicable for numerous dams that do not have advanced hydrological aid.

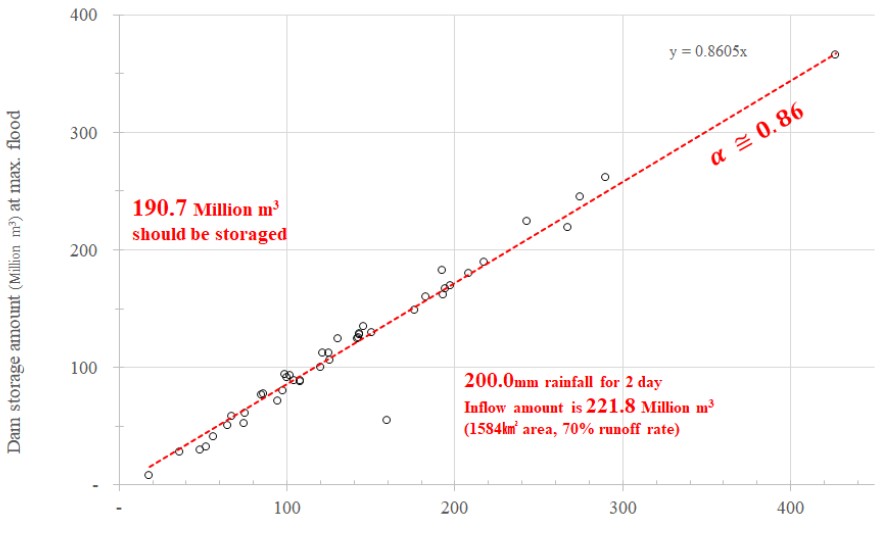

**Figure 5.** Predischarge operation concept at flood event with RSF, red dot line indicats RSF curve and blue dot line represents the process to estimate flood control capacity.

## 4. Discussion

The scatter plots of RSF shown in Figure 4 indicate that each dam has a specific trend between dam storage and flood inflow to the dam. Thus, the RSF method described in Sections 3.2 and 3.3 can be used for estimating predischarge of dam using representative regression equation (for example, red line in Figure 5) of the scatter plot shown in Figure 4. However, for reliable application of the method, it will be necessary to sufficiently consider the RSF characteristics of each dam. The RSF for each dam represented in Figure 4 shows various distributions according to the characteristics of the dam. For example, in the case of

Soyanggang, Hapcheon, Andong, and Yongdam dams in Figure 4a–d, dam storage tends to linearly increase as the inflow increases, and the RSF values range from 0.46 to 0.89. In the contrast, the four dams of Imha, Daecheong, Sumjingang, and Chungju in Figure 4e–h indicate a linearly increasing trend in the case of small inflow, but gradually disperse when the inflow increases above a certain level. In addition, Hwacheon, Namgang, Milyang, and Hoengseong dams in Figure 4i–l do not show a constant trend from the beginning but show various RFS trends according to the flood events. These differences are caused by the characteristics of each dam.

According to the RSF behavior of each dam, this study has classified the RSF trends into three types: Linear, Estranged, and Vague (Figure 6). For example, in the case of the linear type of Figure 6a, as the inflow of the dam increases, the amount of storage in the dam also tends to increase linearly because the dam always has a storage capacity above a certain level compared with the flood inflow. In the case of Soyangang, Hapcheon, Andong, and Yongdam dams, which showed a linear type, they are typically classified as dams that have shown very good performance for flood control.

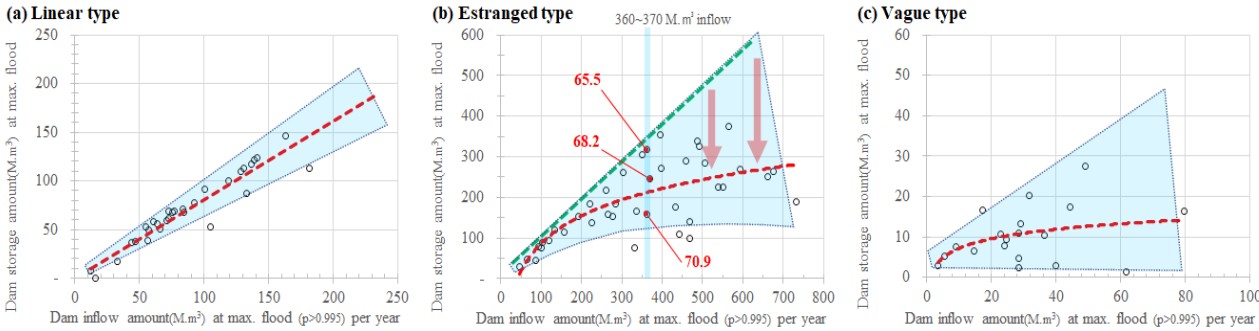

**Figure 6.** Three type of RSF scatter plot: (**a**) Linear type; (**b**) Estrange type; (**c**) Vague type, red dot line indicates average RSF curve, and the green dot line denotes the high threshold line of RSF curve.

In the case of the Estranged type in Figure 6b, a certain portion of the flood inflow is stored similarly to the linear type for a relatively small flood, but the RSF decreases as the inflow increases. This is due to the relatively insufficient storage space of the dam, which does not secure a sufficient level of storage space for the significant (or large) flood inflow and discharges a greater amount downstream. For example, in the case of Daecheong dam in Figure 6b, it shows a constant RSF trend until the inflow for two days reaches about 200 million $m^3$, but the RSF value gradually decreases when the inflow is larger than 200 million $m^3$. In other words, the flood storage efficiency of the Estranged type is poor for the case of flood inflow, which has a certain amount or more. In addition, the RSF trends for the flood inflow of the dam varies depending on the initial storage condition. For example, for all four dams in Figure 4e–h, the linear lines (green dot lines) indicate the high threshold on the RSF points. When the initial water level of the dam is low, a sufficient flood control capacity is secured properly, it is judged that a certain proportion is stored similar to the linear type, and it shows a linear and high-threshold RSF trend. However, when the initial water level of the dam is high, the flood control capacity is insufficient compared to the flood inflow and the RSF seems to decrease. When the high rainfall is expected, dams also secure additional flood control capacity through predischarge, but due to uncertainty in rainfall forecasting, the water level cannot be lowered below the required water supply for the dam [14]; so, it is presumed that RSF would be decreased due to increased flood inflow. In fact, the initial water level of flood events with amount of inflow from 360 to 370 million $m^3$ (see Figure 6b) ranged from EL. 65.5 to 70.9 m, and the higher the initial water level, the lower the RSF values.

In the case of Vague type in Figure 6c, the distribution of annual RSF scatter plots does not show a clear trend. One of the reasons for this result is that it is difficult to find the specific trend due to lack of data or the improper flood control capacity if smaller than that

of the Estranged type; so, the flood control (or storage) capability is greatly affected by the initial water level of the dam. In the case of dams with Vague types including Milyang, Hwacheon, Hoengseong, and Namgang dams, which are usually known as dams that are unfavorable to flood control, insufficient flood control capacity and initial water level are the main causes of improper dam operations [35,36].

As described above, the RSF scatter plot for each target dam was shown and classified into three types according to the RSF behavior of each dam. However, it does not follow clear standards and there are difficulties in practical application. Thus, the features of each type were analyzed for easy and simple classification. In terms of ratio of "watershed area to total storage capacity" and "watershed area to flood control capacity", 12 dams were compared and the results are shown in Table 1.

**Table 1.** The ratio of watershed area, storage, and flood control capacity of each dam.

| Dam | (a) Watershed Area (km$^2$) | (b) Storage Capacity (Million m$^3$) | (c) Flood Control Capacity (Million m$^3$) | (d) Ratio of (a) to (b) | (e) Ratio of (a) to (c) | Type |
|---|---|---|---|---|---|---|
| Soyanggang | 2703 | 2900 | 500 | 0.9 | 5.4 | Linear |
| Chungju | 6648 | 2750 | 616 | **2.4** | 10.8 | Estranged |
| Hwacheon | 3901 | 1018 | 245 | **3.8** | 15.9 | Vague |
| Hoengseong | 209 | 87 | 10 | **2.4** | **22.0** | Vague |
| Namgang | 2285 | 309 | 164 | **7.4** | 13.9 | Vague |
| Hapcheon | 925 | 790 | 80 | 1.2 | 11.6 | Linear |
| Youngdam | 930 | 815 | 137 | 1.1 | 6.8 | Linear |
| Sumjingang | 763 | 466 | 32 | 1.6 | **23.8** | Estranged |
| Daecheong | 3204 | 1490 | 250 | **2.2** | 12.8 | Estranged |
| Milyang | 95 | 74 | 1.4 | 1.3 | **68.1** | Vague |
| Andong | 1584 | 1248 | 110 | 1.3 | 14.4 | Linear |
| Imha | 1361 | 595 | 80 | **2.3** | 17.0 | Estranged |

The results in Table 1 show that the RSF type are related with the ratio of watershed area, storage capacity, and flood control capacity. More specifically, Linear type does not appear when the ratio of watershed area to storage capacity (Table 1) is 2.0 or more, and it is found that the linear type does not appear when the ratio of watershed area to flood control capacity (in Table 1) is 20.0 or more. Thus, if one of the two ratios (Table 1) exceeds the threshold ((d) > 2.0, (e) > 20.0), it shows an Estranged type. If both ratios exceed the threshold (Hoengseong Dam) or one exceeds significantly (Hwacheon, Namgang, Milyang dams), it shows Vague type characteristic. Moreover, the dams that have relatively small storage capacity such as Hoengseong and Milyang seem to have a tendency of going to Vague type easily. For instance, the Imha and the Hoengseong dam have similar (d) and (e) values but they show different types in Table 1. Generally, the flood control behavior becomes rigid easily when the storage capacity of the dams is relatively smaller [35]. Storage capacity of 87 million m$^3$ on the Hoengseong dam seems to be cause, but it needs some further study to validate these relationships. Figure 7 shows the ratio of watershed area to storage (Table 1) and ratio of watershed area to flood control capacity (Table 1) for 12 dams.

The RSF concepts resulted in Section 3.3 and Figure 5 can be applied as simple or filed method for predischarge with heavy rainfall. It can estimate the dam inflow at the time of flooding from the predicted rainfall and it can easily review "the flood control capacity and predischarge required to achieve effective flood control easily", which is shown in Figure 8. For example, assume that 250 mm of rainfall occurs over 2 days in the upstream

area of Chungju dam. Currently, K-Water, which is one of the agencies for dam control in South Korea, considers the runoff rate of the upstream basin of the dam as 70% in monsoon season or heavy rainfall. Thus, if the upstream basin area of Chungju dam is 6648.0 km$^2$ with the runoff rate of 70%, the flood inflow amount can be 1163.4 million m$^3$. Since the RSF type of Chungju Dam is an Estranged type, there are two regression curves: the average RSF curve (red line in Figure 8) and the high-threshold RSF curve (green line in Figure 8). In this case, the capacity for flood control is 536.7 million m$^3$ of the average RSF curve and 800.6 million m$^3$ of the high-threshold RSF curve, and the target water level is 138.6 EL.m and 135.1 EL.m for each RSF curve, respectively. Thus, when 250 mm of heavy rainfall occurs for 2 days, this study suggests that the water level of Chungju dam needs to be controlled to 138.6 m for average flood control capability and to 135.1 m for higher flood control capability compared with the previous year. Based on this result, it can also be used to estimate the target water level for predischarge for flood control in dams when heavy rainfall is forecasted.

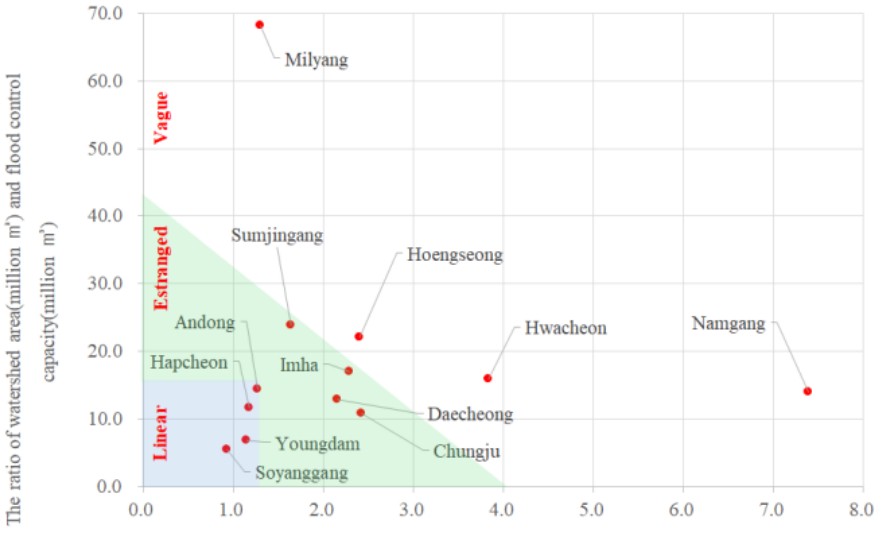

**Figure 7.** Scatter plot of the ratio of watershed area to storage capacity (*X*-axis) and to flood control capacity (*Y*-axis); blue area indicates Linear type, green area shows Estranged type, and other area means Vague type.

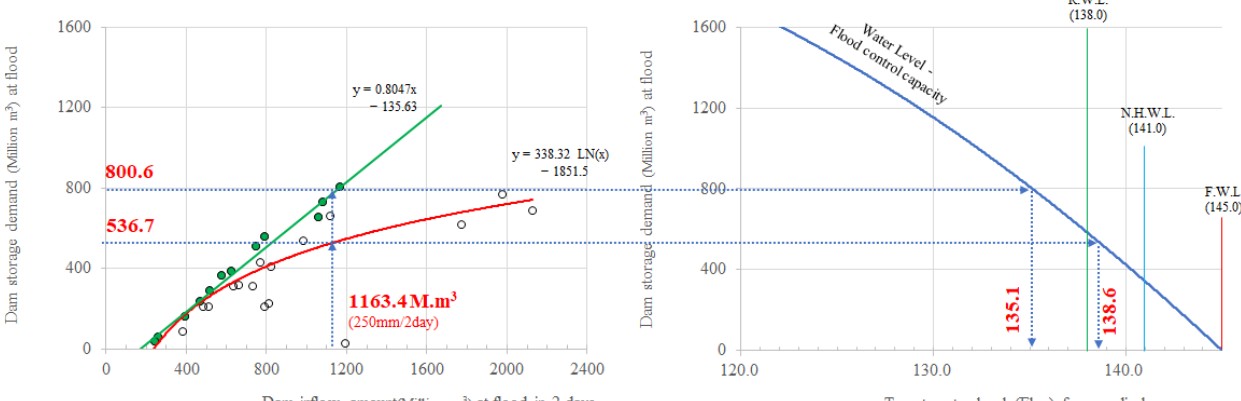

**Figure 8.** Diagram to estimate flood control capacity demand and target water level for predischarge of the Chungju dam; red lined indicats average RSF curve, and green line denotes high threshold line of RSF curve, blue dot line indicates the process to estimate flood control capacity and target water level.

Using the concept, process, and chart presented in this study, it can be used as a reference data for estimating the flood control capacity or predischarge level for dam flood control when heavy rainfall is expected. Thus, it is expected that dam flood control can be performed flexibly if the target water level is determined in advance based on the forecasted rainfall and the water level is lowered through predischarge. In addition, the proposed concept and method will be simple and intuitive way with the dam and reservoir without flood control system. Small-size dams and agricultural reservoirs generally do not have a particular system for flood control, and they usually depend on the experience of the expert in flood event. In this case, the proposed method would be a good alternative to supplement the decision making of the expert. Of course, the proposed method needs an inflow and outflow record of the dam to derive RSF curve. Small-size dams or reservoirs may have relatively lacking or no operation records, and simulated operation records through hydrological modeling could be available. Therefore, further studies attempting to apply the proposed method to small-size reservoirs that do not have operation records are essential. The novelty of this study is that it suggests a simple and intuitive concept and method to estimate the flood control capacity needed to prepare for situations such as heavy rainfall in actual practice; thus, further studies encompassing a wider range of application are needed.

The limitation of this study is that the proposed concept is not an operation method focused on reducing the flood water level, but a predischarge method with the concept of securing flood control capacity. The proposed concept is a method for the dam operator to simply estimate the flood storage and target water level based on rainfall forecast for effective flooding of the dam. Therefore, in order to accurately reduce the flood water level in downstream river, the operation of the dam according to the advanced hydrological analysis is required.

Another limitation is that the proposed method highly depends on rainfall forecasts. Although the target water level for flood control is determined based on the predicted rainfall, if the actual rainfall occurs less or more than expected, valuable water may be ineffectively discharged or, conversely, flood control of the dam may become less-efficient. Therefore, when the proposed method is applied in actual practice, it is necessary to fully consider the uncertainty of the rainfall forecast and the additional safety factors. If precipitation is predicted accurately, the method proposed in this study can be applied as a reasonable standard for effective dam flood control and predischarge.

## 5. Conclusions

This study proposed a novel method to simply estimate the flood control capacity needed to prepare for situations such as heavy rainfall in actual practice. Current dam flood control was mainly performed to reduce the water level of downstream rivers in the event of flood based on hydrological analysis. Sufficient flood control capacity is essential for effective dam flood control; so, it is secured through water level control and predischarge management. The method proposed in this study was applied and analyzed 12 dams in the South Korea; thus, the dam operators will be able to determine the appropriate flood control capacity easily. For this purpose, the daily dam inflow and discharge data from the dams were used as datasets and the frequency matching method was applied to derive a CDF pair with the same return period; then, the RSF concept was suggested and calculated for each dam. Using the RSF values, the characteristics of the dam storage volume at the time of flood inflow were classified into three types, and the required amount of flood control capacity and corresponding target water level were represented. The summary of results are as follows:

(1) To overcome the limitations of quantifying the flood retention effect due to the lag between the inflow and discharge of the dam, the frequency matching method was applied to derive a CDF pair with the same return period, and the RSF concept was analyzed for each dam. For example, as a result of analyzing the RSF value based on the Andong dam,

it can be considered that about 86% of the flood flow of the Andong dam is stored in the dam and only 14% is discharged downstream.

(2) The RSF concept was suggested and applied to the inflows and discharges of 12 major dams in the South Korea, and the RSF regression line representing the flood storage characteristic of each dam was derived. Using the regression line, the storage and discharge of each dam were determined based on the flood inflow volume and a chart for estimating the required flood control capacity was presented accordingly.

(3) By using the trend of the RSF value according to the characteristics of the RSF value for each dam, the storage characteristics were classified into three types: Linear, Estranged, and Vague. In order to quantitatively classify these types, the criteria is presented as a table using the 'ratio of watershed area to storage capacity and 'ratio of watershed area to flood control capacity'. More specifically, Linear type does not appear when the ratio of watershed area to storage capacity is 2.0 or more, and it is found that the linear type does not appear when the ratio of watershed area to flood control capacity is 20.0 or more. Further, when a relatively large flood volume occurs, the RSF is affected according to the initial water level of the dam (Estranged type). Moreover, if both ratios exceed the threshold or one exceeds significantly, it shows Vague type. Finally, it was expected that the proposed methodology in this study could be as a simple way to estimate the flood control capacity or predischarge level for dam flood control, and especially, it also would be a good alternative for the reservoir that does not have a particular flood control system.

**Author Contributions:** Conceptualization, J.K. and H.H.; methodology, J.K., H.H. and D.K.; software, J.J. and H.J.; validation, D.K., J.K. and H.H.; formal analysis, J.J. and H.J.; investigation, D.K. and J.K.; resources, H.S.K.; data curation, H.H. and H.S.K.; writing—original draft preparation, J.K. and H.H.; writing—review and editing, H.H., D.K. and H.S.K.; visualization, J.K. and D.K.; supervision, H.S.K. and D.K.; project administration, H.S.K.; funding acquisition, D.K. All authors have read and agreed to the published version of the manuscript.

**Funding:** This research was supported by research project "Development of future-leading technologies solving water crisis against to water disasters affected by climate change(20220175-001)" funded by the Korea Institute of Civil Engineering and Building Technology (KICT).

**Institutional Review Board Statement:** Not applicable.

**Informed Consent Statement:** Not applicable.

**Data Availability Statement:** The data will be available upon request to the corresponding author.

**Acknowledgments:** This research was supported by research project "Development of future-leading technologies solving water crisis against to water disasters affected by climate change(20220175-001)" funded by the Korea Institute of Civil Engineering and Building Technology (KICT).

**Conflicts of Interest:** The authors declare no conflict of interest.

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
