# Peer review of "Development of Simple Method for Flood Control Capacity Estimation of Dam in South Korea"

_water, doi:10.3390/w14091366_

Round 1

Reviewer 1 Report

This paper suggests a simple methodology to calculate flood control capacity for particular flood events in dams or reservoirs, and its applicability. It seems to have a high level of applicability due to its own simplicity, but it also has to have some revisions as below for more clarity:

  • The authors specified that the objective of the study is the “suggestion of the simple methodology to calculate flood control capacity” In my opinion, the authors seem to have some mistakes in the selection of the study area. All of the studied dams are multi-purpose dams located in South Korea which have their own decision-making system for flood control based on hydrological analysis. When a flood event occurs, they also tried to secure an appropriate amount of flood control capacity based on their flood inflow prediction. Therefore, the authors should have had more discussions about why we have implemented the simple methodology to calculate flood control capacity to the “already well-constructed dams for flood forecasting and decision making”.
  • In the discussion section, the authors described the characteristics of flood storage of the studied dams using the combination of the watershed area, storage capacity, and flood control capacity. Considering the results as shown in Table 1 and Figure 7, it would seem to be persuasive, but not for the Hoengseong dam. It shows different behaviors of flood storage, although it has similar values with another behavior type in Table 1.
  • For instance, the Imha Dam has 2.3 of (d) and 17.0 of (e) item in Table 1, and is considered to be the Estanged type. The Hoengseong dam has 2.4 of (d) and 22.0 of (e) item and is considered to be the Vague type. Do these small changes intend to huge differences between the two dams? The flood control of the dams is also hugely influenced by human control, so it is hard to understand that the behavior changes due to these small changes. The authors should have discussed and to more persuasive explanations for this point.
  • It seems that some of the references are listed in incorrect formats. Please follow the reference rules of the journal.

Reviewer 2 Report

The authors propose a method to control for flooding problems in Korea. This kind of research has strong practical significance, which belongs to the aim scope of Water and should be welcome. Unfortunately, the careless writing of this manuscript is far from meeting the standards of this journal (or any academic journal). The authors should perform the major revisions before reconsidering the publication.

Major concerns:

Such evaluation methods are widely used in flood prevention, and the authors should introduce the situation of other areas.

For the same reason, the novelty of this study needs to be reinterpreted.

Too basic hydrology knowledge does not require extensive statements.

Conclusion needs to be reduced and move parts to the discussion.

This method should have a wider range of application than just a location in Korea.

Although not the most important point, the writing should be more standardized.

Such as:

Line 35, climate change is just one of the many reasons of floods and droughts.

Line 37, the references are clearly erroneous.

Line 86-92, delete it.

Round 2

Reviewer 2 Report

Although some of my concerns are not addressed, from the authors' answers, this is not a realistic task in this manuscript. Therefore, I agree that this manuscript  published in the present format, but I expect the authors  do more intensively work in subsequent studies.